# *Mycobacterium tuberculosis* Small RNA MTS1338 Confers Pathogenic Properties to Non-Pathogenic *Mycobacterium smegmatis*

**DOI:** 10.3390/microorganisms9020414

**Published:** 2021-02-17

**Authors:** Oksana Bychenko, Yulia Skvortsova, Rustam Ziganshin, Artem Grigorov, Leonid Aseev, Albina Ostrik, Arseny Kaprelyants, Elena G. Salina, Tatyana Azhikina

**Affiliations:** 1Shemyakin-Ovchinnikov Institute of Bioorganic Chemistry, Russian Academy of Sciences, 117997 Moscow, Russia; oksana.belonovich@gmail.com (O.B.); ju.skvortsova@gmail.com (Y.S.); rustam.ziganshin@gmail.com (R.Z.); art.grigorov@gmail.com (A.G.); leroymail@gmail.com (L.A.); 2Research Center of Biotechnology, Bach Institute of Biochemistry, 119071 Moscow, Russia; albina.ostrik@gmail.com (A.O.); arseny@inbi.ras.ru (A.K.)

**Keywords:** *Mycobacterium tuberculosis*, *Mycobacterium smegmatis*, small non-coding RNA, MTS1338, proteomics, macrophages, virulence factor

## Abstract

Small non-coding RNAs play a key role in bacterial adaptation to various stresses. *Mycobacterium tuberculosis* small RNA MTS1338 is upregulated during mycobacteria infection of macrophages, suggesting its involvement in the interaction of the pathogen with the host. In this study, we explored the functional effects of MTS1338 by expressing it in non-pathogenic *Mycobacterium smegmatis* that lacks the MTS1338 gene. The results indicated that MTS1338 slowed the growth of the recombinant mycobacteria in culture and increased their survival in RAW 264.7 macrophages, where the MTS1338-expressing strain significantly (*p* < 0.05) reduced the number of mature phagolysosomes and changed the production of cytokines IL-1β, IL-6, IL-10, IL-12, TGF-β, and TNF-α compared to those of the control strain. Proteomic and secretomic profiling of recombinant and control strains revealed differential expression of proteins involved in the synthesis of main cell wall components and in the regulation of iron metabolism (ESX-3 secretion system) and response to hypoxia (*furA, whiB4, phoP*). These effects of MTS1338 expression are characteristic for *M. tuberculosis* during infection, suggesting that in pathogenic mycobacteria MTS1338 plays the role of a virulence factor supporting the residence of *M. tuberculosis* in the host.

## 1. Introduction

Tuberculosis (TB) remains a major threat to global public health, with at least 10 million new cases and 1.2 million deaths in 2019 (World Health Organization’s Global TB Report, 2019, https://www.who.int/tb/publications/global_report/en/ (accessed on 19 February 2021)). The high incidence and mortality rates of TB can be attributed to the capacity of its etiological agent, intracellular bacterium *Mycobacterium tuberculosis* (MTb), to adapt to and survive in the aggressive physiological environment within the host. After being phagocytized by macrophages, mycobacteria are exposed to hypoxic, nitrosative, oxidative, acidic, and nutrient-deficient conditions, but are able to survive by hijacking host defense mechanisms [1,2,3]. Thus, MTb can interfere with maturation of phagosomes into phagolysosomes through a process known as “phagosome maturation arrest” [4,5,6].

Accumulating evidence indicates that mycobacteria can control host defense mechanisms by expressing small regulatory RNAs (sRNAs), which interfere with mRNA translation and stability. These sRNAs are mostly synthesized in response to external stimuli to support bacterial survival and regulate its adaptation to various stresses [7,8]. One such RNA, ncRv11733 (MTS1338), is highly expressed during the MTb stationary phase [9,10], dormancy (a non-replicating state characterized by low metabolic activity) [11] and during infection in the mouse model of tuberculosis [12]. The presence of MTS1338 only in genomes of highly pathogenic mycobacteria and its significant degree of conservation suggests its involvement in MTb virulence. Our recent study revealed that the activation of the host immune system triggered nitric oxide-inducible upregulation of MTS1338 in macrophage-engulfed MTb [12]. Furthermore, MTS1338 overexpression in MTb increased bacterial resistance to the stressful effects of hydrogen peroxide, nitric oxide, acidic environment, and long-term starvation [12,13]. These findings suggest that MTS1338 may promote the survival of mycobacteria in infected macrophages by triggering the transcriptional changes necessary for MTb adaptation to the hostile intracellular environment.

Although the functional activity of bacterial sRNAs has been extensively investigated, there are relatively few such studies in MTb. High genomic CG content and the absence of RNA chaperons (Hfq or ProQ) in mycobacteria suggest the existence of unique activities of their sRNAs in the cell; thus, it is possible that mycobacterial sRNAs might bind enzymes or interact directly with the genome, similarly to some eukaryotic siRNAs [14,15]. For MTS1338, no targets or functional mechanisms have yet been reported.

Therefore, in this study, we examined the functional effects of MTS1338 by expressing it in non-pathogenic *Mycolicibacterium smegmatis* (basonym of *Mycobacterium smegmatis*, MSmeg), which shares about 79% nucleotide sequence identity with MTb and is very similar to it in terms of cell wall composition and metabolism, but lacks the MTS1338 gene [16,17,18,19]. Our results indicate that MTS1338 upregulates the enzymes responsible for the synthesis of the mycobacterial cell envelope and promotes the survival of MSmeg in macrophages, in which the MTS1338-expressing MSmeg induces phagosome maturation arrest and modulates the expression of pro-inflammatory cytokines. These changes are characteristic for pathogenic MTb upon infection, suggesting a role of MTS1338 in MTb virulence.

## 2. Materials and Methods

### 2.1. Bacterial Strains and Growth Conditions

*M. smegmatis* mc(2)155 obtained from the bacterial collection of the Bach Institute of Biochemistry (Research Center of Biotechnology of the Russian Academy of Sciences, Moscow, Russia) was pre-cultured for 24 h at 37 °C on an orbital shaker (200 rpm) in 40 mL of Nutrient Broth (NB) medium (Himedia, Mumbai, India) supplemented with 0.05% Tween-80 and used as inoculums for further experiments. MSmeg recombinant strains were grown in NB supplemented with 50 μg/mL kanamycin (Sigma-Aldrich, St Louis, MI, USA).

### 2.2. Construction of MSmeg Recombinant Strains

The MTS1338 expression plasmid MTS1338-pMV261 was obtained as described previously [11]. The transcription was driven by the rrnB promoter of MSmeg. The eGFP expression plasmid was constructed by inserting the eGFP-encoding gene fragment into BamHI and HindIII sites of pMV261 (Addgene #3668). Then, the expression cassette including the hsp60 promoter, eGFP, and rrnB-T1 terminator was cut from eGFP-pMV261 and introduced into MTS1338-pMV261 “head-to-head” to the MTS1338 expression cassette, yielding the MTS1338-GFP-pMV261 plasmid (Appendix A). Plasmids were amplified in *Escherichia coli* DH5α grown in Luria Bertani (LB) broth and LB agar supplemented with 100 μg/mL ampicillin (Invitrogen, Carlsbad, CA, USA).

MSmeg cells were transformed by electroporation with the MTS1338-pMV261 plasmid and empty pMV261 vector (control), and recombinant strains were selected on NB agar containing 50 μg/mL kanamycin (Sigma-Aldrich). MTS1338 expression was confirmed by quantitative reverse transcriptase-polymerase chain reaction (qRT-PCR). Three independently transformed MTS1338-expressing clones and an equal number of control clones were used in all experiments, which were performed in triplicate.

### 2.3. Infection of Macrophages

RAW 264.7 cells (ATCC^®^ TIB-71™) were cultured in RPMI-1640 (Gibco Europe, Paisley, UK) supplemented with 10% fetal calf serum (FCS) (Gibco) at 37 °C in a 5%-CO_2_ incubator for 24 h until 70~80% confluence.

MSmeg cells grown to the optical density (OD_600_) 0.8 were washed in PBS, resuspended in RPMI-1640 with 10% FCS, and added to macrophages at the multiplicity of infection (MOI) 10:1. Bacterial cells were counted by the micro colony-forming unit (CFU) assay [20] to ensure equal number of control and MTS1338-expressing bacteria used for macrophage infection.

Mycobacterial infection of macrophages was analyzed by the CFU assay. RAW264.7 cells grown in 24-well plates (5 × 10^4^ cells/well) were infected with MSmeg strains for 3 h, washed five times with PBS, then cultured for more 21 h, washed with PBS, and lysed in ice-cold 0.01% sodium dodecyl sulphate (SDS). An aliquot of this suspension was 10-fold serially diluted in solution in milliQ water with 0.05% Tween-80, and 5 μL of each dilution was dropped on LB agar plates in triplicate; the number of colonies was counted after 72 h and expressed as CFU/mL.

### 2.4. RNA Isolation

MSmeg strains transfected with MTS1338-pMV261 (MTS1338) and pMV261 vehicle plasmid (control) were grown in 10 mL of medium to OD_600_ = 0.8 (exponential culture) or OD_600_ = 2.0 (stationary culture). After centrifugation at 4000× *g* for 15 min, pellets were washed twice with fresh medium, rapidly cooled on ice, and centrifuged again. Total RNA was isolated by phenol-chloroform extraction and cell disruption using Bead Beater (BioSpec Products, Bartlesville, OK, USA), as described previously [21], and treated with Turbo DNase (Life Technologies, Carlsbad, CA, USA) to remove traces of genomic DNA.

### 2.5. cDNA Synthesis and qRT-PCR

cDNA was synthesized from 1 mg total RNA using random hexanucleotides and SuperScript III reverse transcriptase (Life Technologies) according to the manufacturer’s protocol. qRT-PCR was performed with specific primers (Appendix A) and qPCRmix-HS SYBR mix (Evrogen, Moscow, Russia) in a LightCycler 480 Real-Time PCR system (Roche, Basel, Switzerland) at the following cycling conditions: 95 °C for 20 s, 61 °C for 20 s, and 72 °C for 30 s, all repeated 40 times. Three biological and nine technical replicates were used to ensure reproducibility; 16S rRNA expression was used for normalization. The results were analyzed with LinRegPCR v 2014.6 [22].

### 2.6. Northern Blotting Analysis

For the detection of MTS1338, 2 μg of total RNA isolated from exponential bacterial cultures was separated on 10% denaturing polyacrylamide gels in 1× TBE buffer and transferred to Hybond N membranes (Amersham, UK). The membranes were hybridized overnight at 42 °C in ULTRAhyb-Oligo hybridization buffer (Life Technologies) with oligonucleotides 1338_NB and 5S_NB, which were 5′-end-radiolabeled (15 pmoles) using 10 μCi of [γ^32^P]-ATP and T4 polynucleotide kinase (Fermentas, Vilnius, Lithuania). After hybridization, the membranes were washed three times in 1× saline-sodium citrate buffer with 0.1% SDS and exposed to X-ray films Retina (Carestream Health, Rochester, NY, USA) to detect radioactivity.

### 2.7. Measurement of Cytokine Production

RAW264.7 macrophages seeded into 6-well plates (Costar, Cambridge, MA, USA) at the density of 7 × 10^5^ cells/well were infected with MSmeg strains at the MOI 10:1 for 3 h, washed five times with PBS, and cultured for the indicated times. RNA extraction and purification from infected macrophages was performed with TRIzol (Evrogen) using the standard phenol-chloroform method. cDNA synthesis and qRT-PCR were performed as described in Section 2.5. Relative mRNA levels were calculated after normalization to β-actin. PCR primers are listed in Appendix A.

To evaluate IL-6 secretion in culture supernatants of infected macrophages, samples were analyzed with the LEGENDplex mouse 8-plex Th1/Th2 panel (BioLegend, San Diego, CA, USA) following the manufacturer’s protocol, and acquired using BD FACSCalibur flow cytometer (BD FACSCalibur, BD Biosciences, San Jose, CA, USA). The data were processed using WinMDI2.8 software (The Scripps Institute, West Lafayette, IN, USA).

### 2.8. Confocal Microscopy

Macrophage infection with mycobacteria was performed as previously described [20] with some modifications. Briefly, RAW 264.7 cells cultured in RPMI-1640 medium with 10% FCS were seeded in antibiotic-free medium on cover glasses (18 × 18 mm Menzel Gläsercoverslips, Thermo Fisher Scientific, Schwerte, Germany) placed in 6-well culture plates (Costar). After 24 h, cells (5 × 10^4^ cells/glass) were infected with MTS1338-GFP strain or GFP-pMV261 (control strain) at MOI 10:1 for 3 h. In the experiments with LysoTracker Red DND-99 (Life Technologies), the dye was added 1 h before the end of infection at the final concentration of 50 nM. After 3 h of infection, the medium was removed and cells were washed three times with PBS, fixed in 1% paraformaldehyde for 10 min, and washed again three times with PBS. For staining with lysosome-associated membrane protein 1 (LAMP-1), cells were incubated with anti-LAMP-1 primary antibodies (sc-20011, Santa Cruz Biotechnology, Santa Cruz, CA, USA) diluted 1:80 in permeabilization buffer (0.1% Triton X-100 in PBS) overnight at 4 °C. Cell monolayers were washed three times for 10 min in the same buffer, incubated with Alexa Fluor 568-conjugated goat anti-mouse IgG (H + L) (1:500; A-21422, Life Technologies) for 1 h at room temperature, and washed three times. Nuclei were stained with 5 µg/mL Hoechst 33,342 (Invitrogen) for 5 min. Mycobacteria were detected by GFP fluorescence at 488 nm, whereas phagosomes (LysoTracker Red DND-99 and LAMP-1 labeling) were visualized at 543 nm, and cell nuclei at 405 nm using an Eclipse TE2000 confocal microscope (Nikon, Tokyo, Japan). The proportion of GFP-labeled phagosomes colocalized with LAMP-1 and LysoTracker Red DND-99 was calculated by analyzing about 50 phagosomes in 4 random fields for each sample. Three independent experiments were performed and the data were statistically analyzed by Student’s *t*-test.

### 2.9. Sample Preparation for Proteomics

MSmeg cultures were grown to OD_600_ = 2.0, centrifuged, resuspended in lysis buffer (100 mM Tris-HCl, pH 7.5, 4% SDS, 10 mM DTT, 1× protease inhibitor cocktail (Sigma-Aldrich)), and disrupted by zirconia beads using a Bead Beater. The resulting lysate was centrifuged at 12,000 rpm for 10 min, filtered through a 20 μm filter, and heated at 85 °C for 10 min. Protein concentration was estimated by the microBCA method (Thermo Fisher Scientific, Waltham, MA, USA). Aliquots containing 50 mg protein were diluted to 1 mg/mL with lysis buffer, and Tris(2-carboxyethyl)phosphine (TCEP) and chloroacetamide (CAA) were added at the final concentrations of 10 and 20 mM for cysteine reduction and alkylation, respectively, performed by heating at 80 °C for 10 min. Proteins were precipitated with five volumes of acetone at −20 °C overnight; pellets were washed twice with acetone, resuspended in 50 mL of 100 mM Tris pH 8.5, 1% (*w*/*v*) SDS by sonication, and treated with trypsin (Promega, Madison, WI, USA) added at the ratio 1/100 (*w/w*, trypsin to protein) for 2 h at 37 °C. Then, the second trypsin portion 1/100 *w*/*w* was added and the sample was incubated overnight at 37 °C. Proteolysis was stopped by 1% TFA and precipitated SDS was removed by centrifugation.

Secreted proteins were purified from supernatants of cultures grown as described above. Supernatants were filtered twice through a 0.2 μm filter and proteins were precipitated with 10% trichloroacetic acid in acetone in the presence of 0.015% sodium deoxycholate.

### 2.10. Liquid Chromatography with Tandem Mass Spectrometry (LC-MS/MS)

LC-MS analysis was carried out in an Ultimate 3000 RSLCnano HPLC system connected to a Q Exactive Plus mass spectrometer (Thermo Fisher Scientific). Protein samples were loaded directly without solid-phase extraction into a trap column (20 × 0.1 mm) packed with Inertsil ODS3 3 mm sorbent (GLSciences, Tokyo, Japan) in the loading buffer (2% acetonitrile, 98% H_2_O, 0.1% trifluoroacetic acid) at the flow rate of 10 mL/min and were separated in a fused silica column (500 × 0.1 mm) packed with Reprosil PUR C18AQ 1.9 (Dr. Maisch, GmbH, Ammerbuch, Germany) into the emitter prepared with P2000 Laser Puller (Sutter Instrument Co., Novato, CA, USA) [23]. Samples were eluted with a linear gradient of solvent A (0.1% formic acid in water) and solvent B (80% acetonitrile, 19.9% H_2_O, 0.1% formic acid) from 4 to 36% of solvent B over 1 h at 0.44 mL/min at room temperature.

MS data were collected in the data-dependent acquisition mode. MS1 parameters were as follows: resolution, 70 K; scan range, 350–2000; max injection time, 50 ms; and automatic gain control target (AGC), 3 × 10^6^. Ions were isolated with a 1.4 m/z window and 0.2 m/z offset targeting the 10 highest-intensity peaks with +2 to +6 charge and 8 × 103 minimum AGC; peptide match was set to preferred, isotope exclusion was enabled, and dynamic exclusion was set to 40 s. MS2 fragmentation was carried out in the higher-energy collision dissociation mode at 17.5 K resolution with 27% normalized collision energy. Ions were accumulated for a maximum of 45 ms with target AGC of 1 × 105. Each sample was analyzed in three biological replicates.

MS raw files were analyzed using PEAKS Studio 8.5 (Bioinformatics Solutions Inc., Waterloo, ON, Canada) [24] and peak lists were searched against UniProtKB/TrEMBLE FASTA (canonical and isoform; version of April 2019) for *M. smegmatis* mc(2)155 with methionine oxidation and asparagine and glutamine deamidation as variable modifications. False discovery rate was set to 0.01 for peptide-spectrum matches and determined by searching a reverse database. Enzyme specificity was set to trypsin in the database search. Peptide identification was performed with an allowed initial precursor mass deviation up to 10 ppm and an allowed fragment mass deviation of 0.05 Da.

The MS proteomics data have been deposited to the ProteomeXchange Consortium via the PRIDE [25] partner repository with the dataset identifiers PXD019813 and 10.6019/PXD019813.

## 3. Results

### 3.1. MTS1338 Inhibited Extracellular Growth of M. smegmatis and Promoted Its Survival in Infected Macrophages

The processed, mature MTS1338 is 108 nt long and has a stable secondary structure with the half-life of 6 h [10]. To evaluate the possible influence of MTS1338 on non-pathogenic mycobacterium, we generated three MSmeg clones expressing the mature form of MTS1338 and compared them with three controls, empty vector-transformed clones. The MTS1338 expression level was checked by qRT-PCR. Intactness of MTS1338 was confirmed by Northern blotting (Appendix A). The results indicated that MTS1338 expression in MSmeg slightly slowed down bacterial proliferation in liquid medium (Figure 1A) and at the same time promoted survival of mycobacteria within RAW264.7 macrophages, as indicated by the growth of the recombinant MTS1338-expressing and control strains isolated from macrophages and plated on solid medium (Figure 1B). An increase in the bacterial survival rate at 24 h post infection (hpi) vs. 3 hpi for both strains could be explained by nonlinear dynamics of bacterial killing in macrophages [26]. In our experiment, MTS1338 expression in macrophages was confirmed by qRT-PCR and appeared to be similar at 3 and 24 hpi (Figure 1C).

### 3.2. MTS1338 Expressing in Non-pathogenic Mycobacterium Reduced the Number of Phagolysosomes in Macrophages

To explore the effect of MTS1338 expression on MSmeg interaction with macrophages, we evaluated the activity of macrophages to engulf mycobacteria in phagolysosomes by analyzing colocalization of the phagosome maturation marker LAMP-1 with GFP-labeled MSmeg. Although phagolysosomes were detected in macrophages infected with both strains (Figure 2A, upper panel), the number of mycobacteria colocalized with phagolysosomes was lower for the MTS1338 strain compared to that of the control strain (Figure 2B). These results indicate that phagosomes containing MTS1338-expressing mycobacteria were less likely to fuse with late endosomes/lysosomes.

To confirm these observations, RAW264.7 cells were labeled with a fluorescent dye (LysoTracker) specific for acidic late endosomes and lysosomes and infected with the two strains. The results revealed a decrease in the colocalization of MTS1338-containing bacteria with phagolysosomes compared to that of the control strain, indicating that MTS1338 expression in mycobacteria prevented phagosome acidification in infected macrophages (Figure 2A, lower panel; Figure 2B).

### 3.3. MTS1338 Affected Cytokine Expression in Infected Macrophages

To further explore the influence of MTS1338 expression in non-pathogenic MSmeg on macrophage immune activity, we analyzed transcription of cytokines IL-1β, IL-4, IL-6, IL-10, IL-12, TGF-β, and TNF-α in RAW264.7 macrophages infected with the MTS1338 and control strains (MOI 10:1) at 4 and 24 hpi. *IL4* transcripts were not detected in any macrophage sample. The expression of *IL1B*, *IL10*, *IL12*, *TGFB,* and *TNFA* mRNA was significantly downregulated in macrophages infected with the MTS1338-expressing strain at both time points (Figure 3A), whereas that of *IL6*, which was almost undetectable in the control strain-infected cells, was markedly increased in the MTS1338 strain-infected cells over the course of infection (by 4-fold compared to that of control) (Figure 3B).

The upregulation of *IL6* production at the transcription level was confirmed by quantification of IL-6 secretion by macrophages into culture medium, which was increased at 24 hpi for RAW264.7 cells infected with the MTS1338-expressing strain (Figure 3C).

### 3.4. MTS1338 Regulated the Expression of Proteins Involved in Iron Metabolism and Cell Wall Remodeling

To clarify the found effects of MTS1338 on MSmeg persistence in macrophages, we performed comparative proteomics of the MTS1338 and control strains. As there is a delay between transcriptional and translational responses in Mycobacteria [27], cultures were analyzed at the stationary phase (OD_600_ = 2.0). In total, 3602 proteins were reliably identified; among them, 109 were unique for the MTS1338 strain and 117 for the control strain (Appendix A).

#### 3.4.1. Proteins Unique for the 1338 Strain

The results indicated that MTS1338 induced the expression of transcription factors and regulators, including LacI, MarR, TetR, GntR, Ethr, DeoR, and WhiB family members, and FurA. WhiB (homolog of WhiB1 in MTb) is a NO-responsive factor, whereas WhiB7 controls expression of genes involved in antibiotic resistance [28]. In MSmeg, WhiB4 and FurA function as peroxide-sensing transcription factors [29].

Furthermore, MTS1338 upregulated SigE, an alternative sigma-factor with an extracellular function shown to be important for MTb physiology and virulence [30,31] as well as anti-sigma-E factor RseA, and induced 2 out of 10 components of the ESX-3 secretion system, EccA3 and EccE3. The other components, EccB3, EccC3, EccD3, and mycP, were detected in proteomes of both strains.

The MTS1338 strain proteome also contained several proteins (I7GAW4, I7GD36, I7FXD2, I7GFT6, heme-thiolate protein P450) belonging to the P450 cytochrome family of heme-thiolate monooxygenases [32] shown to be essential for MTb survival through regulation of sterol metabolism.

Other proteins whose expression was induced by MTS1338 included factors involved in the synthesis of the mycobacterial cell wall, such as polyketide synthase, PKS_KS domain-containing protein, and mycocerosic acid synthase. Mycocerosic acid synthase-like (Mas-like) polyketide synthase, along with acyl-CoA synthase, has been suggested to have a role in the synthesis of branched fatty acids required for lipooligosaccharide production in MSmeg [33,34].

#### 3.4.2. Proteins Absent in the MTS1338 Strain

Among the proteins not detected in the MTS1338 strain, there were many transcription factors (ArsR, GntR, HxlR, IclR, LacI, LuxR, LysR, PadR, TetR, and XRE families) as well as transmembrane proteins with transporter functions, such as sodium and proline co-importer, conserved transmembrane protein, transmembrane ATP-binding protein, ABC transporter I7FV42, integral membrane protein A0QP26, and integral membrane cytochrome D-II CydB I7GAS8. The expression of Sec-independent protein translocase TatC was also not observed in the MTS1338 strain. Mycobacteria utilize the Tat pathway to transport proteins across the cytoplasmic membrane, and Tat mutants have growth defects [35].

Transcriptional regulator PhoP, a component of the PhoP-PhoR transduction signaling system, was found in the control but not in the MTS1338 strain proteome. PhoP is one of the most important participants in the bacterial adaptation to stressful conditions of hypoxia and oxidative stress [36]. It is extremely important for lipid metabolism, in particular for the synthesis of polyketide-derived lipids [37].

### 3.5. MTS1338 Promoted Secretion of Proteins Involved in Bacterial Virulence and Cell Wall Permeability

Mycobacteria are known to secrete a large number of proteins that modulate immune response of the host organism; therefore, to determine whether MTS1338 affected secretion in MSmeg, we compared secretomes of the MTS1338 and control strains.

#### 3.5.1. Proteins Unique for the MTS1338 Strain Secretome

Among 10 proteins secreted only by the MTS1338 strain, 6 had homologs in MTb (Table 1); of them, all belonged to the functional category “cell wall and cell processes”. The following proteins had the highest representation in the MTS1338 secretome:

SGNH_hydro domain-containing protein is a homolog of the MTb enzyme Rv0518 belonging to the GDSL lipase superfamily of lipolytic enzymes. It has been shown that *Rv0518* expression in MSmeg changes cell morphology and composition of cell wall lipids; increases the content of total lipids and trehalose dimycolates; confers resistance to environmental, intracellular, and antibiotic stresses; and enhances infection ability and intracellular survival of mycobacteria [38].

Dacb2 (I7FBC5) is a putative D-alanyl-D-alanine carboxypeptidase; these enzymes play a vital role in peptidoglycan cross-linking and regulate the expression of surface glycopeptidolipids, which could mask phosphatidyl-myo-inositol mannosides from recognition by Toll-like receptor 2 [39].

A putative septum form domain-containing protein that has an MTb homologue encoded by the *Rv3835* gene. It was annotated as ZipA, a protein stabilizing a Z-ring [40], which acts as an FtsZ assembly factor [41] and is involved in remodeling of the mycobacterial peptidoglycan cell wall at the division site.

The secreted protein A0R445 containing a transglycosylase domain is a homolog of the MTb protein RpfA, which promotes resuscitation and growth of dormant, non-growing mycobacterial cells [42,43].

Antigen MTB48 (EspB_PE domain-containing protein) is homologous to ESX-1 protein EspB in MTb, which is responsible for host cell death and may promote virulence [44].

#### 3.5.2. Proteins Absent in the MTS1338 Strain Secretome

Among the proteins whose secretions were blocked in the MTS1338 strain (Table 2), two are very interesting.

Secreted antigen 85-C FbpC is a member of the antigen 85 complex (Ag85) responsible for the affinity of mycobacteria to fibronectin and possessing mycolyltransferase activity [45]. MSmeg has five putative mycolyltransferases: FbpA, FbpB, FbpC, and two homologs of FbpD [46], which have some functional redundancy [47]. We found that while both control and MTS1338 strains secreted FbpA, FbpB, and low amounts of FbpDs, FbpC secretion was inhibited in the MTS1338-expressing strain.

Porin MspA is the main porin in MSmeg; it forms a water-filled channel that favors permeation of hydrophilic molecules such as cations, amino acids, Fe^3+^, and phosphates [48]. As porin-mediated influx of nutrients is a major determinant of MSmeg growth [49], the inhibition of MspA secretion by MTS1338 may account, at least in part, for the slower proliferation of the MTS1338 strain (Figure 1A).

### 3.6. Validation of the Proteomic Results by qRT-PCR

To confirm the differential expression of proteins of particular interest, we analyzed their transcription in MTS1338 and control strains. To reveal the dynamics of transcription, we performed qRT-PCR for both exponential (OD_600_ = 0.8) and stationary (OD_600_ = 2.0) mycobacterial cultures.

We checked the transcription levels of *MSMEG_0965*, *MSMEG_5872*, and *MSMEG_3886* coding for porin MspA, transcriptional regulator PhoP, and translocase TatC of the Tat secretion system, respectively (Figure 4A–C). The proteins coded by these genes were found as expressed in the control strain. The mRNA levels of the three genes were lower in the MTS1338-expressing strain and those of *MSMEG_0965* and *MSMEG_3886* were downregulated at both exponential and stationary phases. Analysis of two genes, *MSMEG_2433* (Figure 4D) and *MSMEG_0615* (Figure 4E), whose products (D-alanyl-D-alanine carboxypeptidase Dacb2 and ATPase of ESX3 secretion system EccA3, respectively) were unique for the MTS1338 strain, confirmed differential expression of *MSMEG_0615* in the stationary phase.

## 4. Discussion

Currently, sRNAs are widely recognized as a distinct group of bacterial virulence factors that play a central role in the regulation of gene expression and adaptation to stresses. MTS1338 is a non-coding sRNA expressed by pathogenic mycobacteria, which is upregulated during infection in vivo and in vitro in activated macrophages [12]. In this study, we showed that MTS1338 changed the phenotype of non-pathogenic MSmeg normally not expressing this sRNA, so that the mycobacteria acquired the ability to delay phagosome maturation in macrophages, which promoted its intracellular survival, indicating adaptation to stressful conditions and defense against host immune response.

Importantly, MTS1338 expression in MSmeg resulted in changes of the cytokine expression profile in infected macrophages. The transcription of cytokines TNF-α, TGF-β, IL-1β, IL-6, IL-10, and IL-12, which is either absent or negligible in non-infected macrophages (Broad Institute Cancer Cell Line Encyclopedia, https://portals.broadinstitute.org/ccle (accessed on 19 February 2021); Human Protein Atlas, https://www.proteinatlas.org/ (accessed on 19 February 2021)), is strongly upregulated in mycobacteria-infected macrophages [50,51]. However, the upregulation of TNF-α, IL-1β, and IL-12 in macrophages infected by MTb is lower than in those infected by MSmeg [52], suggesting that the immune response is attenuated by pathogenic MTb. Our results showed that in RAW264.7 macrophages infected by the MTS1338 strain, the expression of TNF-α, IL-1β, and IL-12 was significantly downregulated compared with that of macrophages infected by the control strain (Figure 3A), indicating that MTS1338-expressing mycobacteria inhibited the production of pro-inflammatory mediators in macrophages.

In contrast, the expression of IL-6 is increased in MTb-infected compared to MSmeg-infected cells [52], and we observed that IL-6 was upregulated in macrophages infected with MTS1338-expressing MSmeg compared to those infected with the control strain (Figure 3B,C). IL-6, a pleiotropic cytokine produced by a variety of cells, including macrophages, is considered to play a pro-inflammatory function in TB and to be important for control of MTb infection [53]. However, Ernst et al. (2003) [54] have proposed a novel, bacteria-oriented role for IL-6; they showed that IL-6 secreted by MTb-infected macrophages selectively inhibited a subset of IFN-γ-responsive genes and thus may contribute to the inability of the cellular immune response to eradicate infection. By modulating the balance of inflammatory cytokines secreted by the host in response to MTb infection, MTS1338 may favor bacterial survival and, consequently, influence the outcome of infection.

Among the changes caused by MTS1338 in the protein expression profile of MSmeg, the most important indicated remodeling of the bacterial cell wall. The unique composition of the mycobacterial cell wall, which consists of three main covalently linked components: peptidoglycan, arabinogalactan, and mycolic acids [55,56,57], is essential for the intracellular survival of mycobacteria in host cells. Comparison of proteomes and secretomes of the control and MTS1338-expressing strains revealed differential expression of proteins involved in the synthesis of all major cell wall components. Thus, D-alanyl-D-alanine carboxypeptidase Dacb2, RpfA homologue, and septum form domain-containing proteins (which play vital roles in peptidoglycan synthesis and cross-linking), appeared in the MTS1338 secretome, whereas mycolyltransferase FbpC disappeared. The loss of FbpC increases the ratio of trehalose dimycolate (TDM, cord factor) to mycolated arabinogalactan [47,58]. TDM is an important structure necessary for maintaining cell wall integrity, and its content in the mycobacterial cell envelope was shown to be relatively high in the early stages of MTb infection [56]. Lipooligosaccharides, another group of trehalose-based lipids present in the mycobacterial envelope, may also be increased in MTS1338-expressing MSmeg as evidenced by the upregulation of Mas-like polyketide synthase involved in lipooligosaccharide production. As the cell envelope is at the forefront of mycobacteria interaction with the host, its structure and response to immune factors are important in determining the state of infection. A peptidoglycan-derived dipeptide has been shown to stimulate both innate and adaptive immunity [59], whereas TDM is necessary to block phagosome–lysosome fusion, providing mycobacterial growth in macrophages and inducing pro-inflammatory cytokines [60,61,62,63,64]. Hence, changes in the cell wall biosynthesis induced by the MTS1338 strain are consistent the inhibition of phagolysosome maturation and shifts in inflammatory cytokines transcription in the recombinant strain.

The secretion system of MSmeg was also significantly affected by MTS1338. Thus, the major MSmeg porin, MspA, involved in the export of Fe^3+^ and some other essential hydrophilic solutes, was absent in the 1338 secretome. Previous reports indicate that deletion of porins in mycobacteria improve their persistence in eukaryotic cells [65], and that MspA increases the susceptibility of MTb to β-lactam antibiotics [66], suggesting that MspA downregulation should enhance mycobacterial survival in the host. In addition, TatC, a component of the Tat secretion system involved in the export of β-lactamases that provides natural resistance of mycobacteria to β-lactam antibiotics and is considered to play a role in MTb pathogenesis [67], was absent in the 1338 proteome. Further studies are necessary to determine how MTS1338 influences the resistance of mycobacteria to antibiotics.

Another important cluster of proteins involved in mycobacterial secretion and induced by MTS1338 are those of the ESX-3 secretion system. ESX-3 is essential for iron homeostasis and survival of mycobacteria in vivo [68], which is consistent with the significance of iron uptake for MTb virulence revealed in animal models [3,69] and patients with TB [70]. Iron is an essential element for MTb, which expresses over 40 enzymes requiring iron as a cofactor. The ESX-3 system contributes to siderophore-mediated iron acquisition [71]. Furthermore, ESX-3 is involved in heme utilization by MTb [68], which is necessary for the function of many transcriptional regulators and cytochromes present in mycobacteria.

In conclusion, our results indicate that *M. tuberculosis* sRNA MTS1338 confers to non-pathogenic MSmeg the properties characteristic for pathogenic MTb upon infection, suggesting MTS1338 as a virulence factor of MTb. However, the underlying mechanisms need further investigation. The survival-promoting effects of MTS1338 could be linked to the arrest of phagolysosome maturation caused by MTS1338-expressing mycobacteria, which demonstrated bacterial cell wall rearrangement and changes in iron metabolism. The increased persistence in macrophages together with the downregulation of pro-inflammatory mediators due to MTS1338 expression could be important indicators of suppressed host immune response, which may affect the outcome of MTb infection.

## Figures and Tables

**Figure 1 microorganisms-09-00414-f001:**
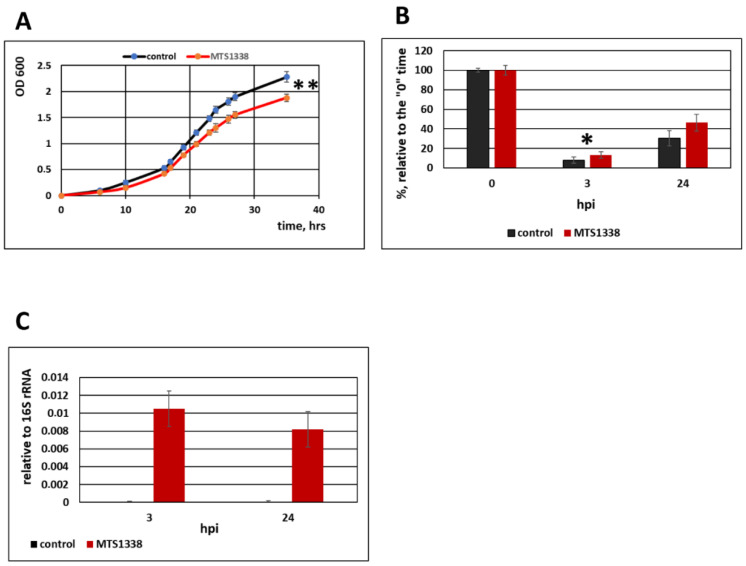
MTS1338 regulated the growth and survival of mycobacteria. (**A**) Growth of control (pMV261-transformed) and MTS1338-expressing strains in Nutrient Broth medium supplemented with 0.05% Tween-80. (**B**) Survival of control and MTS1338 strains in RAW264.7 macrophages infected at multiplicity of infection (MOI) 10:1. Colony forming units (CFUs) were counted at 3 and 24 h post infection (hpi), and the results presented relative to time “0” taken as 100%. (**C**) MTS1338 expression in RAW264.7 macrophages infected by MTS1338-expressing MSmeg was analyzed by qRT-PCR at the indicated time post-infection. The data are presented as the mean ± SD of three independent experiments: * *p* < 0.05 and ** *p* < 0.01. ND, non detected.

**Figure 2 microorganisms-09-00414-f002:**
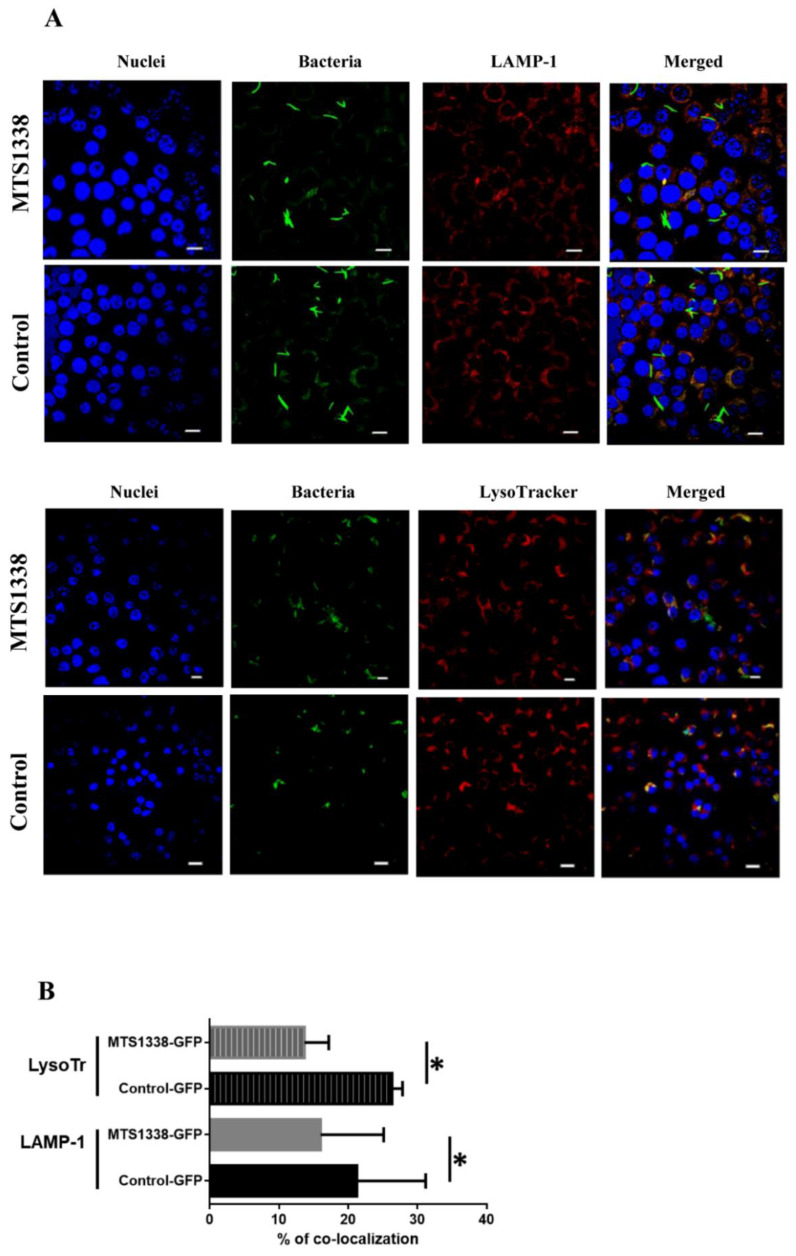
MTS1338-expressing MSmeg inhibited phagosome maturation in RAW264.7 macrophages. (**A**) Representative fluorescence images of RAW 264.7 cells infected with MSmeg-GFP and MSmeg-GFP-MTS1338 for 6 h. Green, MSmeg; red, LAMP-1 or LysoTracker; blue, nuclei. Colocalization of bacteria with LAMP-1 or LysoTracker is indicated in orange. (**B**) Quantitative analysis of mycobacteria colocalization with LAMP-1 and LysoTracker. The data are presented as the mean ± SD of about 600 mycobacterial cells analyzed per each sample: * *p* < 0.05. Scale bars, 10 μm.

**Figure 3 microorganisms-09-00414-f003:**
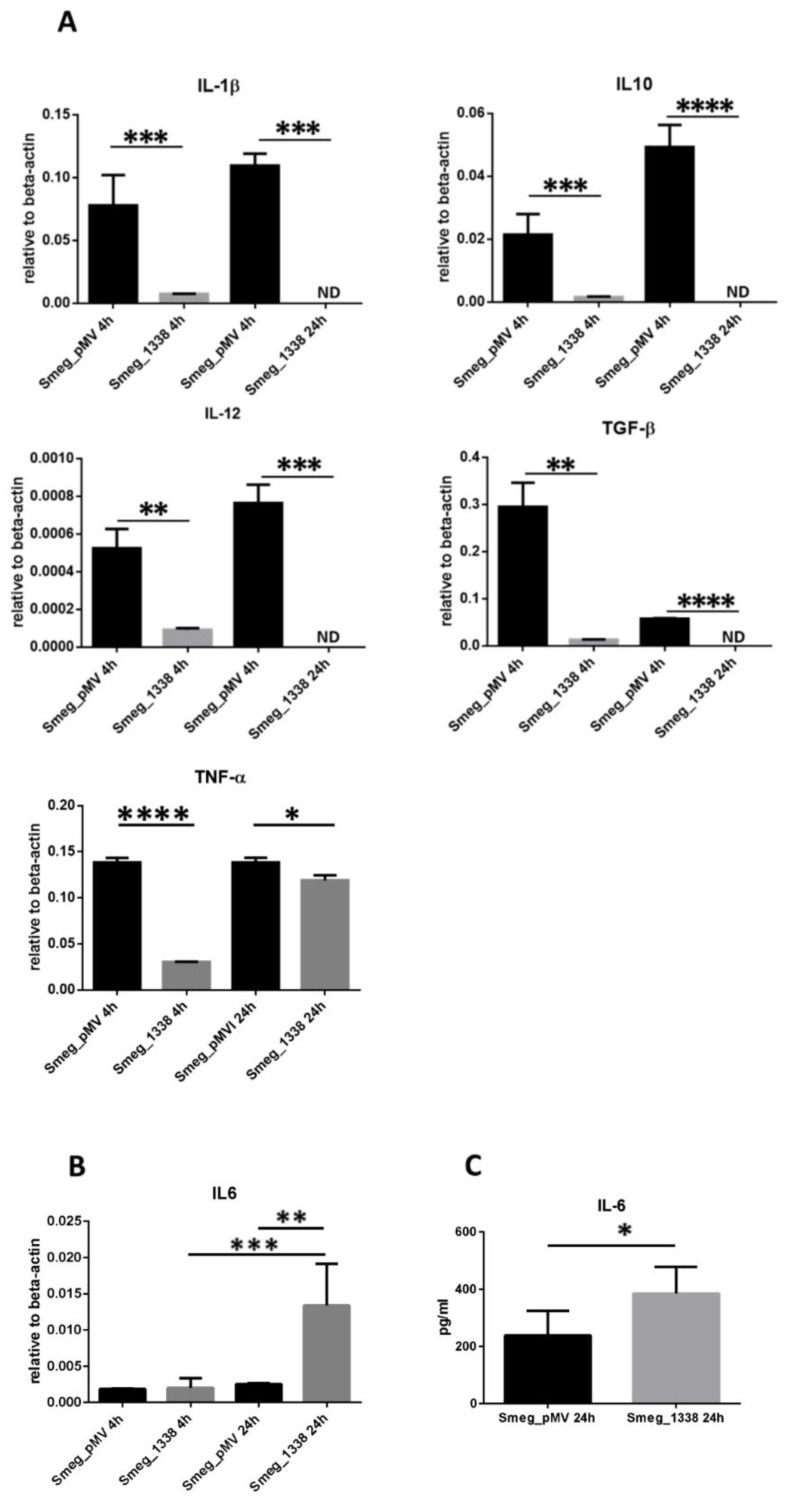
(**A**,**B**) Expression of cytokines in macrophages infected with MTS1338 and control strains. RAW264.7 macrophages were infected with control and MTS1338-expressing mycobacteria and analyzed for mRNA expression at 4 and 24 hpi. (**C**) Secretion of IL-6 by infected RAW264.7 cells at 24 hpi. The data are presented as the mean ± SD of three independent experiments: * *p* < 0.05, ** *p* < 0.01, *** *p* < 0.001, and **** *p* < 0.0001 (by Student’s *t*-test). ND, not detected.

**Figure 4 microorganisms-09-00414-f004:**
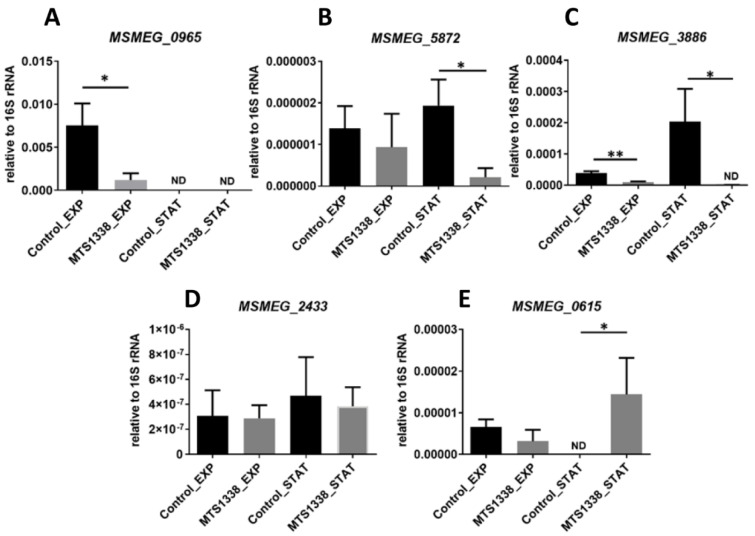
Confirmation of differential expression by qRT-PCR. mRNA expression of *MSMEG_0965* (**A**), *MSMEG_5872* (**B**), *MSMEG_3886* (**C**), *MSMEG_2433* (**D**), *MSMEG_0615* (**E**) was determined in exponential (EXP) and stationary (STAT) cultures and normalized to that of 16S rRNA. * *p* < 0.05, ** *p* < 0.01.

**Table 1 microorganisms-09-00414-t001:** Secreted proteins unique to the MTS1338 secretome.

Accession, UniProt	Unique Peptides, Average	Protein	Gene	MTb Ortholog	Functional Category According to Mycobrowser
A0QY05	7	SGNH_hydro domain-containing protein	*MSMEG_3489*	*Rv0518*	Cell wall and cell processes
A0QV35	3,7	Putative D-alanyl-D-alanine carboxypeptidase Dacb2 (Penicillin-binding protein)	*MSMEG_2433*	*Rv2911*	Cell wall and cell processes
A0QRU0	3	Uncharacterized protein	*MSMEG_1237*	no	
A0R773	2,7	Sugar ABC transporter substrate-binding protein	*MSMEG_6804*	no	
A0QQ67	2,7	GH16 domain-containing protein	*MSMEG_0645*	no	
A0R639	2,3	Septum_form domain-containing protein	*MSMEG_6414*	*Rv3835*	Cell wall and cell processes
A0R445	2,3	Secreted protein	*MSMEG_5700*	*Rv0867c*	Cell wall and cell processes
A0QQY6	2,3	Uncharacterized protein	*MSMEG_0921*	*Rv0477*	Cell wall and cell processes
A0QP20	2,3	MHB domain-containing protein	*MSMEG_0243*	no	
A0QNK4	2,3	Antigen MTB48 ESX-1 secreted protein B PE domain	*MSMEG_0076*	*Rv3881c*	Cell wall and cell processes

**Table 2 microorganisms-09-00414-t002:** Secreted proteins unique to the control secretome.

Accession, UniProt	Unique Peptides, Average	Protein	Gene	MTb Ortholog	Functional Category According to Mycobrowser
I7G354	5,0	Secreted antigen 85-C FbpC	*MSMEG_3580*	*Rv0129c*	Lipid metabolism
A0QR29	3,3	Porin MspA	*MSMEG_0965*	no	
A0QRM0	3	UPF0234 protein	*MSMEG_1165*	*Rv0566c*	Conserved hypotheticals
A0R4A7	3	DUF732 domain-containing protein	*MSMEG_5766*	no	
A0QV51	2,7	Methylmalonate-semialdehyde dehydrogenase	*MSMEG_2449*	no	
A0R061	2,3	HesB/YadR/YfhF family protein	*MSMEG_4272*	*Rv2204c*	Conserved hypotheticals

## Data Availability

The MS proteomics data have been deposited to the ProteomeXchange Consortium via the PRIDE partner repository with the dataset identifiers PXD019813 and 10.6019/PXD019813.

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
