# Peer review of "Mycobacterium tuberculosis Small RNA MTS1338 Confers Pathogenic Properties to Non-Pathogenic Mycobacterium smegmatis"

_microorganisms, 2021, doi:10.3390/microorganisms9020414_

Round 1

Reviewer 1 Report

The study presents the analysis of the overexpression of Mycobacterium tuberculosis (MTb) small RNA MTS1338 (that it is upregulated during Mtb infection in vivo and in activated macrophages) in non-pathogenic Mycobacterium smegmatis (MSmeg) that lacks the MTS1338 gene. In the transformed strain a number of functional parameters were analyzed: the growth in culture, the survival in macrophages, production of mature phagolysosomes, changes in the production of cytokines, and a proteomic and secretomic profiling of the recombinant strain. Methods are well described and the results are consistent with the idea that sRNA MTS1338 is a virulence factor of MTb, as the changes are those characteristic for pathogenic MTb upon infection. However, the underlying mechanisms need further analysis. 

As correctly described and discussed, the secretion system of MSmeg was significantly affected by overexpression of MTS1338. Two of the proteins involved are of particular interest, MspA increases the susceptibility of MTb to ß-lactam antibiotics, and TatC, involved in the export of ß-lactamases, provide natural resistance of mycobacteria to ß-lactam antibiotics. One simple analysis that should be done is just to measure the MIC of some antibiotics for the recombinant MSmeg compare with wild type.

On the other hand, discussion is well performed and very convincing.

Author Response

Reviewer 1

One simple analysis that should be done is just to measure the MIC of some antibiotics for the recombinant MSmeg compare with wild type.

RE: We thank the reviewer for the idea of measurement the MIC of some antibiotics for the recombinant and control M.smegmatis strains. However, we suppose that this experiment will not affect our main conclusions that MTS1338 plays a role of a virulence factor supporting the residence of M. tuberculosis in the host. We are planning to determine how MTS1338 influences the resistance mycobacteria to antibiotics in our further studies.

Reviewer 2 Report

I consider the said study very well done and timely, given the fact we need more targets in MTB. The identification of the small RNA in previous studies was rounded out nicely by expressing it in a non pathogenic model. They demonstrated survival in macrophages and reduction in metabolism of cells expressing the said small RNA. The proteomics studies threw up several interesting enzymes related to lipid and cell wall metabolism in Mycobacteria. The authors further showed transcriptomics results to back up the proteomics for some of the candidates. They also showed the localization of the bacteria expressing the RNA inside macrophages using fluorescence microscopy. In summary, this study was well designed and executed, providing evidence through multiple approaches. It will add to the increasing number of targets in MTB and hopefully help tackle increasing drug resistance. Future studies can pick up the aspect of how this small RNA when expressed leads to drug resistance.

Please perform minor spell check. 

There is mention of porin-mediated transport of nutrients in the Msmeg strain used. Is there any difference in B12 transport related genes? If so, please mention.   

Author Response

Reviewer 2

There is mention of porin-mediated transport of nutrients in the Msmeg strain used. Is there any difference in B12 transport related genes? If so, please mention.  

RE: We thank the reviewer for this question. Indeed, many studies demonstrated the ability of mycobacteria to use exogenous B12 (Kipkorir et al, De Novo Cobalamin Biosynthesis, Transport and Assimilation and Cobalamin-Mediated Regulation of Methionine Biosynthesis in Mycobacterium smegmatis. J Bacteriol. 2021 doi: 10.1128/JB.00620-20), however little is known about proteins involved in mycobacterial B12 transport. It was reported about some ABC transporters (Gopinath Krishnamoorthy et al, A vitamin B12 transporter in Mycobacterium tuberculosis, 2013, Open Biol. 120175 http://doi.org/10.1098/rsob.120175).

We carefully examined the list of proteins changing their expression in response to MTS1338, but no candidate proteins associated with the transport of vitamin B12 were found.